# AgentHPO: Large Language Model Agent for Hyper-Parameter Optimization

Siyi Liu[1], Chen Gao[2]*, Yong Li[2]*

[1]Hong Kong University of Science and Technology (Guangzhou)
[2]Tsinghua University

ssui.liu1022@gmail.com, chgao96@gmail.com, liyong07@tsinghua.edu.cn

Hyperparameter optimization is critical in modern machine learning, requiring expert knowledge, numerous trials, and high computational and human resources. Despite the advancements in Automated Machine Learning (AutoML), challenges in terms of trial efficiency, setup complexity, and interoperability still persist. To address these issues, we introduce a novel paradigm leveraging Large Language Models (LLMs) to automate hyperparameter optimization across diverse machine learning tasks, which is named AgentHPO (short for LLM Agent-based Hyperparameter Optimization). Specifically, AgentHPO processes the task information autonomously, conducts experiments with specific hyperparameters (HPs), and iteratively optimizes them based on historical trials. This human-like optimization process largely reduces the number of required trials, simplifies the setup process, and enhances interpretability and user trust, compared to traditional AutoML methods. Extensive empirical experiments conducted on 12 representative machine-learning tasks indicate that AgentHPO not only matches but also often surpasses the best human trials in terms of performance while simultaneously providing explainable results. Further analysis sheds light on the strategies employed by the LLM in optimizing these tasks, highlighting its effectiveness and adaptability in various scenarios.

## 1. Introduction

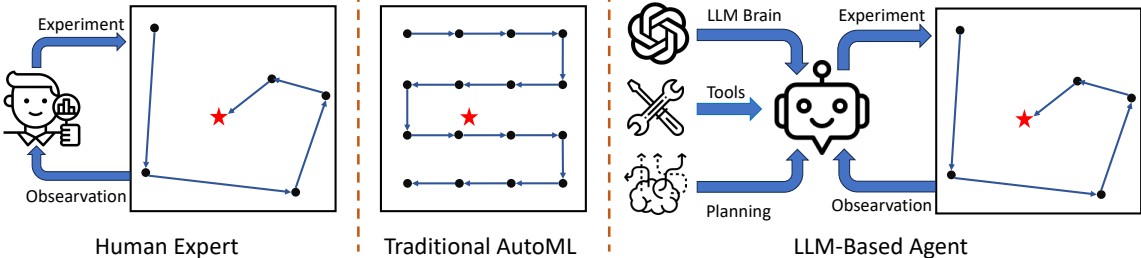

Figure 1: Comparative Frameworks in Hyperparameter Optimization: Human Expertise, Traditional AutoML, and LLM-Based Agents

In Machine Learning (ML), Hyperparameter Optimization (HPO) is indispensable for fitting models to diverse problems. This process involves adjusting hyperparameters (HPs) that shape the model's structure and learning method, which are set before training and greatly affect performance [1, 2]. Traditionally, human experts with algorithmic knowledge play an essential role in HPO, leveraging their theoretical and practical ML expertise to refine models for improved performance. However, the complexity of HPO, due to the extensive range of configurations and task-specific demands, makes it a time-intensive process heavily reliant on an expert's ability to adapt their knowledge to new scenarios [3–5].

---

*Corresponding Authors.

Second Conference on Parsimony and Learning (CPAL 2025).

To alleviate the intensive labor of manual HPO, the ML community has turned towards Automated Machine Learning (AutoML) [6]. AutoML frameworks employ methods like Bayesian optimization [7] to explore the HP space, reducing the need for extensive human intervention. Despite showing promise, AutoML-based HPO still faces the following drawbacks: **Time-Intensive Trials**: AutoML's reliance on numerous trials for black-box optimization is effective but burdensome, particularly with complex tasks and large datasets. The balance between the number of trials and computational demand creates a trade-off between efficiency and the quality of results [3, 4, 8]. **Complex Setup**: Despite AutoML's versatility across domains and hardware, its setup is intricate. That is, it involves choosing suitable tools and defining optimal HP spaces, where misconfigurations can lead to inefficiency or poor performance without experts' supervision [9]. **Lack of Interpretability**: The lack of transparency in many AutoML methods leads to concerns about their dependability. It is crucial, particularly for less experienced users, to have a clear understanding of how different HPs impact the model and the reasoning behind specific configuration choices. This interpretability gap often makes manual tuning a more trusted choice over AutoML [10–12].

In this work, we propose **AgentHPO**, which utilizes the advancements in Large Language Models (LLMs)-powered autonomous agents, to overcome the complexities faced by traditional AutoML methods. As illustrated in Figure 1, AgentHPO draws on the extensive domain knowledge, advanced tool utilization, and sophisticated reasoning of LLMs to ease the dependence on human experts.

To be specific, AgentHPO is innovatively designed with two specialized agents: *Creator* and *Executor*. The *Creator* agent acts as the starting point of optimization, enabling users to input task-specific details, such as dataset characteristics, model structure, and optimization goals, in a natural language format. This agent adeptly interprets the input and generates initial HPs, emulating the expertise of a human specialist. Subsequently, based on the HPs provided by the *Creator*, the *Executor* agent takes on the responsibilities of training models, recording experimental data, and conducting outcome analyses. The *Creator* uses insights from the *Executor*'s training history to iteratively refine the HPs, thereby streamlining the optimization process and making it more intuitive and efficient. With the above designs, AgentHPO effectively addresses several known challenges in traditional AutoML methods, as follows:

**High Trial Efficiency**: By leveraging the specialized capabilities of the *Creator* and *Executor*, AgentHPO significantly reduces the time and resources required for conducting multiple trials.

**Simplified Setup and Configuration**: AgentHPO's natural language input feature makes it easier to input task-specific details and effectively define optimal HP search spaces. This significantly reduces the complexity and likelihood of misconfiguration.

**Improved Interpretability and Trust**: AgentHPO's clear, textual explanations of HP choices foster greater user trust and understanding, making this approach more accessible and preferable, particularly for those without expert-level knowledge of HPO.

The key contributions of our work can be summarized as follows:

- To the best of our knowledge, we take the first step to introduce LLM-based autonomous agents in HPO problems. Our investigation sheds light on the extensive capabilities and adaptability of LLMs in automating and optimizing ML processes.

- We propose an LLM agent-based general framework comprised of distinct and specialized agents: *Creator* and *Executor*. The two agents work collaboratively to assist a wide range of users, especially those without extensive expertise, efficiently tuning ML models.

- We carried out extensive experiments on 12 representative ML HPO tasks across various domains and the results showcase the method's practicality and superior performance.

## 2. Related Works

### 2.1. LLM-based Autonomous Agents

Large language models (LLMs) have emerged as a pivotal element in AI agent development, prized for their extensive knowledge bases, reasoning and planning capabilities, generalization potential, and adeptness at tool use [13, 14]. The integration of LLMs as the core cognitive component in these agents has paved the way for their versatile application across various real-world domains. For instance, MetaGPT [15] has leveraged LLM-based multi-agent systems for collaborative software development tasks. Park et al. [16] explored the use of agents for simulating intricate human interactions. Voyager [17] crafted an agent capable of navigating the complex environment of the Minecraft game. Further pushing the boundaries, Boiko et al. [18] introduced Coscientist, an initiative harnessing the power of LLM-based agents for pioneering autonomous chemical research. In this paper, we delve deeper into the capabilities of LLM-based autonomous agents in the AutoML field, focusing on addressing HPO.

### 2.2. LLMs for AutoML

Large Language Models (LLMs) have the potential to significantly enhance ML tasks by autonomously decomposing and executing complex ML operations. These models are being increasingly recognized for their ability to deliver convenient, comprehensive, and reliable decision-making across a variety of applications and tasks. For instance, AutoML-GPT [19] leverages LLMs to conduct HPO by iteratively prompting with data and model cards, along with mimicking model training via LLMs. Similarly, MLcopilot [20] utilizes LLMs, informed by past experiences and knowledge, to predict optimal HP settings in a categorized manner. CAAFE [21] employs LLMs for automated feature engineering in tabular data to generate semantically meaningful features. EvoPrompting [22] integrates LLMs as adaptive operators in an evolutionary neural architecture search (NAS) algorithm. Auto$^2$Graph [23] deploys LLM-based agents to devise tailored solutions for diverse graph-structured data and learning tasks. Moreover, MLAgentBench [24] introduced a suite of ML tasks specifically for benchmarking AI research agents, with an emphasis on advancing research in the ML domain. However, these methods are either not specifically designed to address HPO or lack a mechanism to iteratively refine HPs based on direct, empirical evidence from historical training performance. Distinct from previous research, our AgentHPO introduces the first agent-based task-agnostic HPO framework, uniquely designed to iteratively optimize HPs across various real-world ML tasks.

### 2.3. LLMs-enhanced Model Optimization

Recent studies reveal the potential of LLMs in optimization tasks involving trajectory input. OptFormer [25] trains a transformer model on extensive collections of HPO data to predict new HP configurations. OPRO [26] performs optimization by prompting LLMs with solution-score pairs, and Zhang et al. [27] extended this strategy to HPO settings. However, these works overlook the impact of training logs and necessitate manual code configuration adjustments and executions. That is, these works only partly use LLMs as an assistant tool, which still highly relies on huge human efforts. In contrast, our research focuses on LLM-based autonomous agents that incorporate detailed training logs into experimental documentation. This approach leads to a more efficient HPO framework, significantly reducing human involvement.

## 3. Methodology

Figure 2 and Algorithm 1 illustrate our AgentHPO framework, which streamlines the HPO process. Initially, users provide their dataset characteristics and learning goals in natural language, offering a more user-friendly alternative to traditional, code-intensive configurations. The process commences with an LLM-empowered *Creator* agent $C$ that interprets the user-provided task-specific background information. This agent then generates an initial HP configuration. Subsequently, *Executor* agent $E$

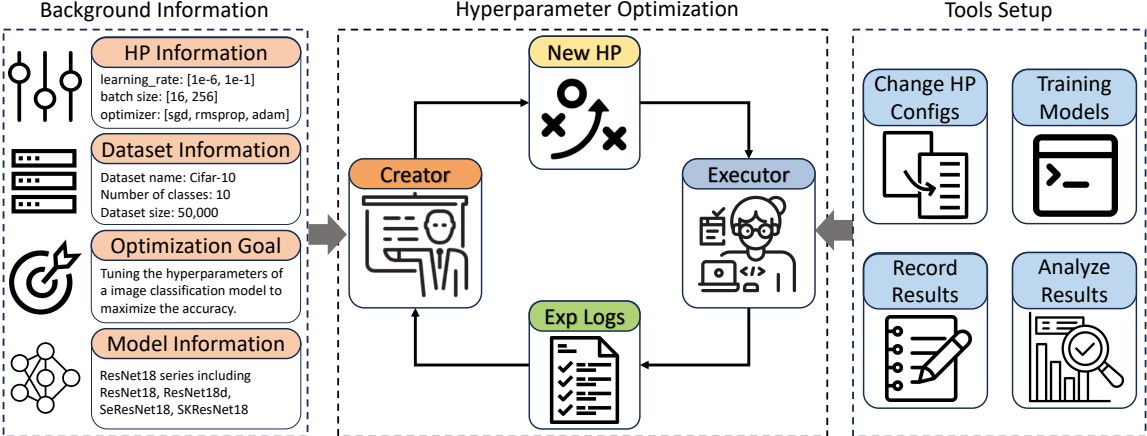

Figure 2: Overview of our AgentHPO. The AgentHPO processes textual background information, autonomously conducts experiments with specific HPs, and iteratively optimizes them. This human-like optimization process enables AgentHPO to achieve high performance with minimal trials and provides users with an interpretable optimization solution.

employs this configuration to train models, analyze the training outputs, and log the experimental data. Leveraging the accumulated training history, the *Creator* agent iteratively refines and proposes new HPs. This approach streamlines the execution of various ML HPO tasks, significantly reducing the necessity for deep AutoML expertise or high-level coding skills. Subsequent sections delve into the mechanisms by which LLM agents execute HPO, utilizing the provided information and historical training logs.

## 3.1. Creator Agent

This section describes our methodology for prompting the *Creator* agent for initial HP generation and subsequent optimization. The designed prompts enable the LLM to not only generate appropriate HPs for a specific ML task but also to iteratively refine them. The prompt structure comprises several critical elements, each contributing to the informed decision-making process of the *Creator* agent:

- **HP Information**: This supplies the agent with a foundational understanding of the HPs that require optimization, encompassing a list of HP names and their descriptions. Additionally, value ranges are specified to constrain the LLM's search space.

- **Dataset Information**: This includes essential statistical details of the dataset such as the number of samples, feature dimensions, and number of target classes.

- **Optimization Goal**: This defines the objective that the *Creator* agent aims to achieve, which may involve maximizing or minimizing specific metrics. These metrics pertain to model efficacy as well as operational constraints like memory usage or training duration.

- **Model Information**: This pertains to fundamental details about the training model, including its architecture and the number of parameters it contains.

The composite of these components constitutes the background information $\mathcal{B}$, equipping the *Creator* agent with the insights needed for strategic and informed HP generation. With this setup, the *Creator* agent, denoted as $C = init(LLM, \mathcal{B})$, is well-equipped to commence the HP generation and optimization tasks. The complete prompts utilized by the *Creator* agent are provided in Appendix A.5

## 3.2. Executor Agent

The role of the *Executor* agent commences upon receipt of the generated HP configurations from the *Creator* agent. Tasked with the crucial responsibility of conducting experiments, the *Executor* assesses the effectiveness of these HP settings. Each training session under the *Executor* is conceptualized as an interactive environment, allowing the agent to execute specific actions and observe their outcomes. The *Executor* agent utilizes a comprehensive suite of tools $\mathcal{T}$ to facilitate these actions:

- **Change HP Configs**: The agent is equipped to modify the HPs in response to newly updated configurations.
- **Training Models**: The agent possesses the capability to execute model training scripts, allowing for evaluation of the outcomes of the altered configurations.
- **Analyze Results**: Upon the completion of model training, the agent scrutinizes the training logs, which include the trajectory of training and validation metrics, to conduct a comprehensive analysis of the training outputs and synthesize a summary of the experiment.
- **Record Results**: Finally, the agent documents the outcomes of the training and the corresponding analyses in the experimental logs for future reference.

Equipped with these tools, the *Executor* agent is empowered to not only implement and adjust HPs but also to critically evaluate model performance and systematically record the findings. This capability ensures a structured and methodical approach to experimental ML workflows. The *Executor* is instantiated as $E = init(LLM, \mathcal{T})$, ready to undertake its designated tasks. Detailed prompts for the *Executor* agent are available in Appendix A.6.

## 3.3. Iterative Hyperparameter Optimization

To enhance model performance through HPO, expert practitioners typically consult historical experimental records to deduce potential avenues for improvement. This iterative process, which tests new HP configurations $H_t$ and validates them through experimentation, aims to converge on an optimized model performance $H^*$. Emulating this expert approach, the *Creator* ($C$) and *Executor* ($E$) agents operate within a similar paradigm for HP optimization in our AgentHPO framework.

Within each iteration $t$ of the optimization process, the *Creator* agent $C$ generates a set of HPs $H_t$, along with the rationales $R_t$ for their selection, by analyzing the accumulated experimental logs $\mathcal{L}$ (step 8 in Algorithm 1). The *Executor* agent $E$ then takes these HPs $H_t$ and carries out the experiments, with the outcomes of these experiments being captured in $L_t$ (step 9 in Algorithm 1). The results $L_t$ encompass the performance metrics and a comprehensive analysis post-experimentation. These findings are appended to the experimental logs $\mathcal{L}$, creating a historical record that includes the HPs $H_t$, their explanations $R_t$, and the experimental results $L_t$ (step 10 in Algorithm 1). Thus, the experimental logs $\mathcal{L}$ (depicted in Figure 6) serve as a dynamic repository, documenting the iterative progress and informing the *Creator* agent's subsequent decisions. In this context, $\mathcal{L}$ can also be viewed as a memory block, archiving sequences of the agents' past observations, reflections, and actions. This repository harnesses prior experiences to inform future strategy formulation and decision-making processes within the AgentHPO framework.

This cyclical refinement, driven by the *Creator* agent's analysis and the *Executor* agent's experimental results, ensures a progressively optimized set of HPs $H_t$ and systematically steers the process towards the ideal HP $H^*$.

## 3.4. Explainable Hyperparameter Optimization

As we discussed in Section 3.3, experimental logs $\mathcal{L}$ encompass not only the HPO trials but also provide comprehensive explanations for each trial. Therefore, our AgentHPO addresses the prevalent issue of interpretability in HPO processes by providing optimal HP $H^*$ and its corresponding reasoning $R^*$. Furthermore, the *Creator* agent offers an in-depth final analysis upon the conclusion

of each experiment (step 12 in Algorithm 1). These analyses are pivotal as they furnish users with a summary of the HPO process, enhancing their understanding of the impact of various HPs on the model's performance. Finally, the *Creator* agent can also propose potential avenues for future optimization, thereby guiding users toward more effective and efficient HPO strategies. For more details, please refer to Appendix 5.4.

# 4. Benchmark Setting

Table 1: Comprehensive overview of tasks, datasets, and models, datasets marked with $^\dagger$ indicate their release occurred post the knowledge cutoff dates of GPT-3.5 and GPT-4. For all metrics, higher is preferable.

| Task | Sub-Task | Dataset | Model | Metrics |
|------|----------|---------|-------|---------|
| CV | Image classification | Cifar-10 [28] Butterfly Image$^\dagger$ [30] | ResNet-18 [29] | Accuracy |
| | Segmentation | CityScapes [31] | ENet [32] | IOU |
| NLP | Text classification | Ecommerce Text$^\dagger$ [33] SST2 [35] | DistilBERT [34] | Accuracy |
| | Machine Translation | Opus Books [36] | T5-Small [37] | BLEU |
| RecSys | Matrix Factorization CTR | MovieLens 1M [38] | LightGCN [39] DeepFM [40] | NDCG@10 AUC |
| Tabular | Classification Regression | Water Portability$^\dagger$ [41] House Price$^\dagger$ [43] | XGBoost [42] | F1 Score $R^2$ Score |
| GNN | Node classification Link prediction | Cora [44] Pubmed [44] | GCN [45] VGAE [46] | Accuracy AUC |

## 4.1. Task descriptions

In this paper, our methodology is applied across a diverse array of 12 tasks, covering disciplines such as Computer Vision (CV), Natural Language Processing (NLP), Recommender Systems (RecSys), Tabular Data, and Graph Neural Networks (GNN). The specifics of these tasks are presented in Table 1. Our task selection includes both classic datasets and recent challenges from Kaggle, ensuring that the study is representative of both traditional benchmarks and current, real-world problems, which lie outside the scope of the language models' pre-training data[2]. This range was meticulously selected to span a broad spectrum of complexities and contemporary relevance. A comprehensive discussion on each task and the corresponding HP search spaces can be found in Appendix A.3 and Table 2.

## 4.2. Experimental Setup

**Evaluation of AgentHPO**. Our experimental procedure entails conducting 10 trials per run ($T = 10$). At each trial milestone $t$ (specifically at the $\{1, 3, 5, 10\}$ trial marks), we record the best metric performance achieved among the first $t$ trial results. This tiered evaluation process allows us to assess performance improvements throughout the trials.

**Baseline Settings**. In our experiments, we implemented **Random Search** and **Bayesian Optimization** [7] as baseline methods, with each executing 10 runs per experiment, totaling 100 iterations. The same process of recording the best metric performance at each trial milestone is applied to baselines, ensuring a consistent and fair comparison. Given that a random search with 100 trials exhibits a 99% probability of locating a near-optimal HP region, constituting merely 5% of the search grid [47], the peak performance across these trials suggests the optimal outcome achievable via human-directed

---

[2]Based on the information provided by OpenAI's official documentation, the GPT-3.5 model encompasses knowledge up to September 2021, while the GPT-4 model includes updates up to April 2023.

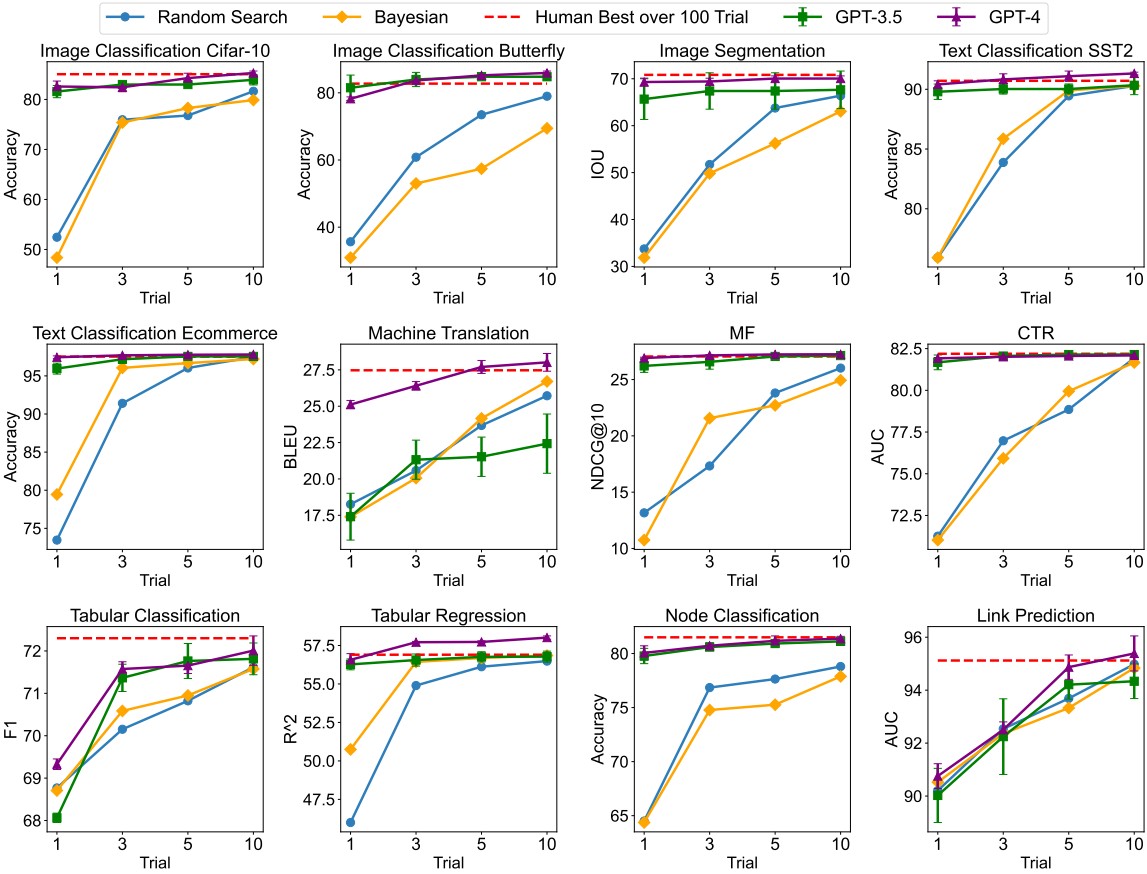

Figure 3: Performance trajectory of various baselines across trials, with the X-axis indicating the trial count and the Y-axis showing the associated task metrics. To benchmark performance, we showcase the optimal outcome within 100 trials as a representation of the highest achievement attainable by human effort.

efforts (denoted as **Human Best**). For the LLM-based optimization baseline, we adopted **OPRO** [26] by only recording the HP-score pairs in experimental logs, while keeping other settings consistent with AgentHPO.

**AgentHPO Settings**.  In our study, the AgentHPO framework incorporates OpenAI's GPT-4 and GPT-3.5 as LLMs. The APIs for GPT-4 and GPT-3.5 are set as `gpt-4-1106-preview` and `gpt-3.5-turbo-1106` respectively. Due to the higher operational costs associated with GPT-4, we strategically conduct 5 runs using GPT-4, compared to 10 runs for GPT-3.5. For the *Creator* agent within AgentHPO, the temperature parameter is set to 1 to enhance exploration, while other HPs remain at default settings. The AgentHPO is implemented based on LangChain's API `zero-shot-react-description` for both agents.

# 5. Results and Analysis

## 5.1. Trajectory over Trails

Figure 3 delineates the performance trajectories for a suite of tasks over a series of trials, with detailed numerical results presented in Table 3. The key observations from this study are summarized as follows:

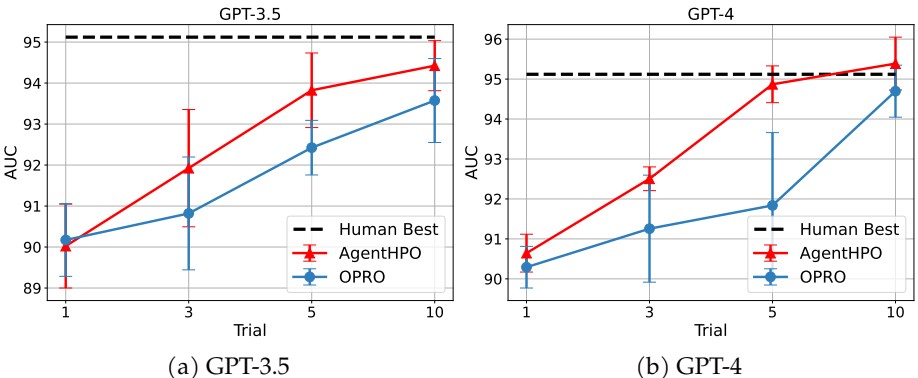

|  |  |
|:---:|:---:|
| (a) GPT-3.5 | (b) GPT-4 |

Figure 4: Link Prediction performance trajectory comparison between OPRO and AgentHPO

- **Superior Performance**: AgentHPO consistently outperforms random search baselines and, in some instances, surpasses human best results. Specifically, in the 10th trial ($T = 10$), AgentHPO's GPT-3.5 model exhibits a $3.83\%$ average improvement over random search results, though it is slightly lower than the best human results by $1.18\%$. Meanwhile, GPT-4 showcases a remarkable $6.66\%$ average enhancement over random search and a $1.52\%$ average improvement over the best human performances. These results confirm the AgentHPO's proficiency in leveraging intrinsic knowledge for HPO.

- **Initial Trial Efficiency**: Both GPT-3.5 and GPT-4 exhibit impressive performance in the initial trials ($T = 1$). For example, GPT-3.5 demonstrates a notable $56.81\%$ average improvement over random search in the first trial, while GPT-4 achieves an even more impressive $61.29\%$. These figures highlight the models' ability to effectively utilize pre-learned knowledge for swift and effective optimization right from the start, underscoring their initial trial proficiency.

- **Robustness on New Datasets**: AgentHPO shows remarkable effectiveness on newer datasets that were released after their training cut-off. For example, on the Butterfly dataset, GPT-4 achieves $85.92 \pm 0.57\%$, surpassing the human benchmark of $78.27\%$. Similarly, GPT-3.5 also demonstrates strong performance on the Butterfly dataset, achieving $84.79 \pm 1.01\%$ in its 10th trial. These results highlight the models' ability to employ broad optimization strategies, showcasing their adaptability and a comprehensive understanding of optimization principles, enabling them to perform effectively on both familiar and new datasets.

- **GPT-4's Superiority Over GPT-3.5**: A notable finding in our analysis is the consistent outperformance of GPT-4 over GPT-3.5. On average, in initial trials across all tasks, GPT-4 surpasses GPT-3.5 by $4.65\%$, demonstrating its enhanced efficiency in initial stages of optimization. This trend continues into later stages, with GPT-4 maintaining a $3.08\%$ higher performance than GPT-3.5 by the 10th trial. Additionally, GPT-4 exhibits more robust results, evidenced by its lower average standard deviation of $0.358$ compared to GPT-3.5's $0.994$. These statistics underscore GPT-4's superior optimization capability and its consistency in delivering more reliable and effective results.

Collectively, these findings elucidate the advanced capabilities of AgentHPO in the realm of HPO, showcasing the potential of LLMs in the field of AutoML.

## 5.2. Influence of Experimental Logs

Experimental logs play a crucial role in AgentHPO, recording historical HP training outcomes, such as training/validation losses and evaluation metrics that evolve over epochs (see Figure 6 and Appendix A.7 for examples). To verify how training logs assist the *Creator* agent in generating better HPs, we compared our AgentHPO with OPRO, which only records solution-score pairs (in the HPO setting, HP-score pairs) in logs. We conducted experiments on link prediction tasks as the performance improvement over trials is more significant than with other tasks.

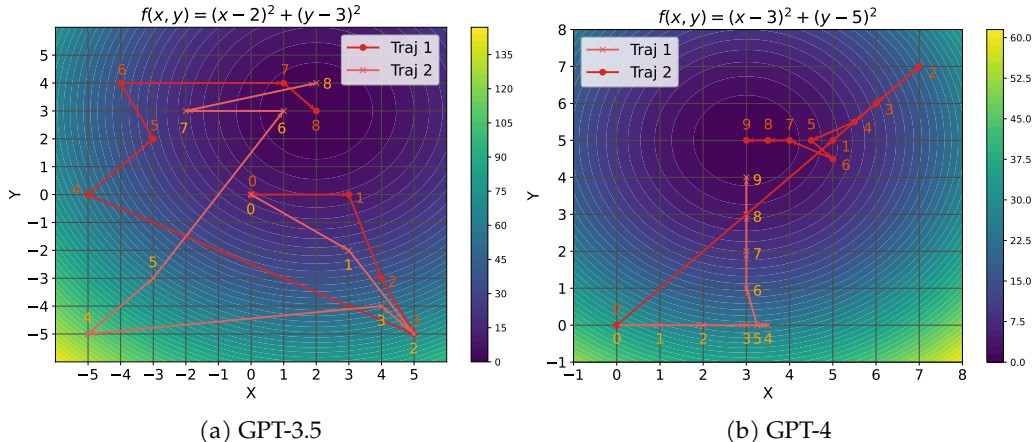

Figure 5: Comparison of optimization trajectories between GPT-3.5 and GPT-4

As illustrated in Figures 4a and 4b, our AgentHPO achieves better optimization over trials compared to OPRO. These results further demonstrate the importance of experimental logs in helping the *Creator* agent detect training patterns more effectively (e.g., over- or under-fitting and converge speeds), thus generating more suitable subsequent HPs.

## 5.3. Optimization Strategy Analysis

To elucidate the underlying optimization strategy employed by AgentHPO, we embarked on a task to optimize a convex function to identify its minimum value over two variables. Specifically, we tasked GPT-3.5 with minimizing the function $f(x, y) = (x - 2)^2 + (y - 3)^2$, with $x, y \in [-5, 5]$, and assigned GPT-4 a similar function $f(x, y) = (x - 3)^2 + (y - 5)^2$, with $x, y \in [-10, 10]$. These functions were chosen to test the agents' ability to search for the optimal $x$ and $y$ values within a given range, a non-trivial challenge for the LLMs due to the absence of explicit boundary values and function information[3].

The optimization trajectories, visualized in Figures 5a and 5b, reveal distinct behaviors for each model. Notably, both LLMs initiated their search from the central region of the defined space, aligning with the heuristic that the middle is a logical starting point without prior data. Subsequently, GPT-3.5 exhibited a search pattern akin to a "random search" strategy, later refining its search to progressively converge on the function's minimum. This suggests that GPT-3.5's strategy may involve an "educated" random search, leveraging accumulated information to hone in on the target. The strategic search patterns of GPT-4, shown in Figure 5b, highlight a strong link between model performance and optimization path. Trajectory 1 demonstrates a systematic approach akin to heuristic or gradient-descent methods, quickly identifying and following a promising direction toward the function's minimum. This suggests GPT-4's rapid recognition and focused pursuit of an optimal path, reflecting a deep understanding of optimization landscapes. Conversely, Trajectory 2 exhibits an initial broad exploration, resembling a global search strategy, before zeroing in on the minimum through a refined local search, similar to simulated annealing or Bayesian optimization techniques.

In conclusion, the behavior of AgentHPO affirms the viability of LLMs as powerful tools in the HPO. AgentHPO can substantially reduce the time and computational resources typically required for HPO while simultaneously increasing the probability of achieving near-optimal solutions.

---

[3]The range of values for $x$ and $y$ has been deliberately narrowed to more clearly illustrate the trajectories generated by GPT-3.5. This adjustment helps mitigate potential confusion in the plot that could arise from the inherent randomness of the model's search strategy.

## 5.4. Explainable Hyperparameter Optimization

In the domain of HPO, the interpretability of model decision-making processes is of paramount importance. For this reason, we have showcased a segment of the experimental logs from AgentHPO in Figure 6, with a more detailed example available in Appendix A.7. These logs offer not only historical performance data but also enhance the process's interpretability. They allow users to monitor the progression of model training, thereby promoting transparency in HPO and providing an explainable HPO solution.

Despite this commonality, the two models exhibit notable differences in their approach to generating HP configs. GPT-4 distinguishes itself by providing more detailed explanations for each HP's reasoning. It goes beyond mere logging of training progress. As seen in Experimental 4 in GPT-4's logs, it delves into the rationale behind each parameter choice, drawing on the results from previous experiments to inform its decisions. This capability suggests a more advanced understanding of the optimization landscape, allowing GPT-4 to strategically deduce HP values that are likely to yield improvements in model performance. Conversely, GPT-3.5's approach within AgentHPO resembles that of an educated guesswork system. While it can effectively generate new HP sets and provide a degree of rationale for its choices, its ability to reason and iterate based on historical performance data is less sophisticated compared to GPT-4. The GPT-3.5-based agent relies more heavily on established heuristics and incremental adjustments, which, although effective, may not capture the full complexity of the optimization process as adeptly as GPT-4.

The nuanced distinction between the two models' strategies underscores the evolution of LLMs and their potential to enhance HPO. GPT-4's nuanced reasoning and learning from past results represent a significant step forward, offering a more strategic and potentially more effective pathway to optimal HP configurations.

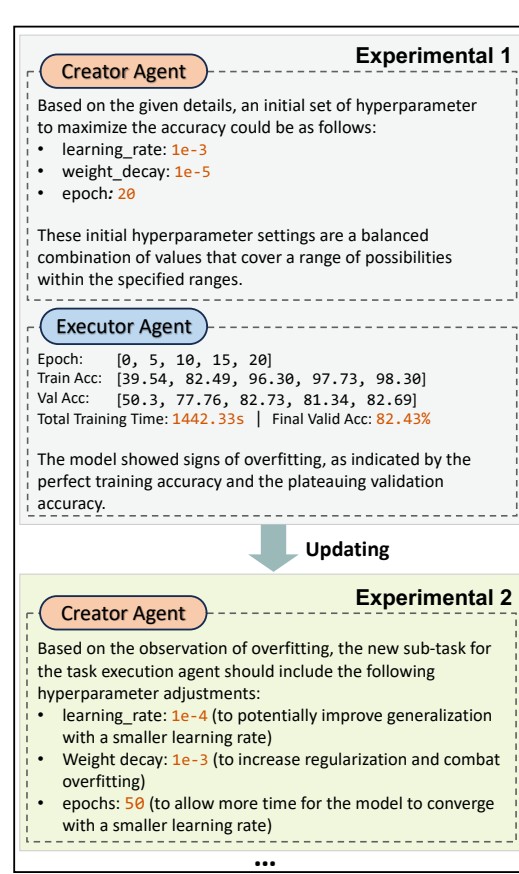

Figure 6: Example of experimental logs.

## 6. Conclusion

In this work, we take the pioneering step in exploring and replacing human efforts in tuning machine-learning models with large language model-based agents. We propose a creator-executor framework that shows superior performance compared with human trials and baseline methods, demonstrating a promising research direction that eases human labor in machine learning tasks. For future work, we aim to enhance the benchmark by incorporating more sophisticated AutoML baselines for comparison.

## Acknowledgements

This work was supported by the National Key Research and Development Program of China under Grant No. 2022YFB3104702, and by the National Natural Science Foundation of China under Grant Nos. 72442026, 62272262, and 72342032.

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

# A. Appendix

## A.1. Algorithm of AgentHPO

We present the Algorithm of our proposed AgentHPO in Algorithm 1

---

**Algorithm 1** The optimization algorithm of AgentHPO

---

1: **Input:** Background Information $\mathcal{B}$ in natural language. Tools $\mathcal{T}$
2: **Output:** Optimized hyperparameters $H^*$, Experimental logs $\mathcal{L}$
3: Initialize *Creator* $C = init(LLM, \mathcal{B})$.
4: Initialize *Executor* $E = init(LLM, \mathcal{T})$.
5: Initialize Experimental Logs $\mathcal{L} = []$.
6: Set number of trials $T$
7: **for** $t = 1$ to $T$ **do**
8:     $H_t, R_t \leftarrow C.create(\mathcal{L})$
9:     $L_t \leftarrow E.execute(H_t)$
10:     $\mathcal{L}.append([H_t, R_t, L_t])$
11: **end for**
12: $H^*, \mathcal{L} \leftarrow C.analyze(\mathcal{L})$
13: **return** $H^*, \mathcal{L}$

---

## A.2. Evaluating AgentHPO's Optimization Capabilities on Novel Models

To further assess the effectiveness of AgentHPO on novel models, we examine its capability to optimize emerging architectures developed after the LLM knowledge cutoff. We conducted experiments on the KAN model [48], introduced in April 2024, which differs significantly from traditional MLPs and requires specialized hyperparameters and optimization strategies. In the binary classification setting of KAN, we used a synthetic dataset generated by scikit-learn for our experiments. The results are shown in Figure 7:

As shown, AgentHPO produces strong initial trial results, outperforming both random search and Bayesian optimization by 18.8%. Specifically, at T=10, AgentHPO (GPT-3.5) outperforms random search and Bayesian optimization by 2.65% and 1.39%, respectively. Moreover, AgentHPO (GPT-3.5) achieves a 1.81% improvement over OPRO (GPT-3.5), with similar trends observed for GPT-4. These results demonstrate that AgentHPO effectively optimizes hyperparameters for novel models.

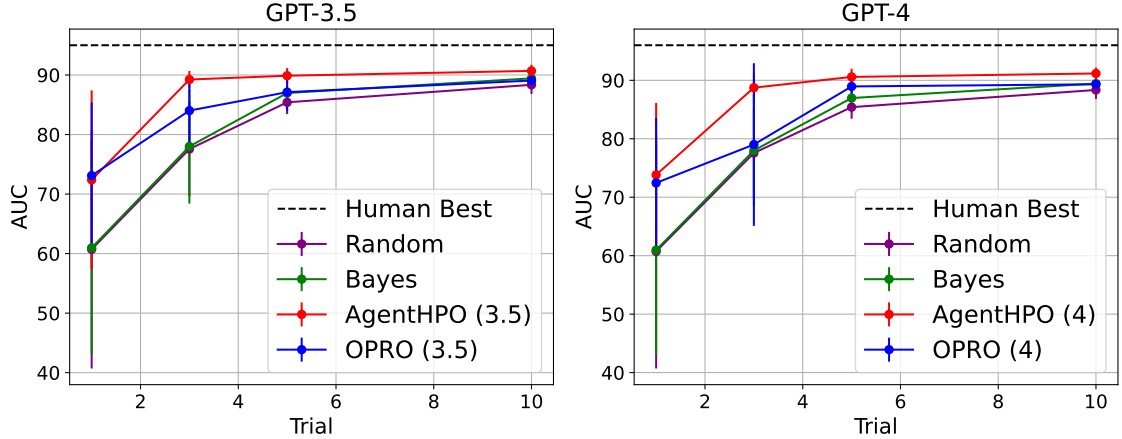

Figure 7: KAN model performance trajectory comparison

## A.3. Detailed Task Description

For organizational clarity and ease of analysis, these tasks have been categorized into distinct groups as follows:

**Computer Vision**: In this domain, we concentrate on two primary sub-tasks: Image Classification and Image Segmentation. For Image Classification, we use ResNet-18 [29] with accuracy as the performance metric. We conduct HPO on two datasets, the first is the well-known Cifar-10 [28], and the second is the newer Butterfly Image dataset [30] from Kaggle, which was released after the LLM release date. For Image Segmentation, our experiments utilize the CityScapes dataset [31] with ENet [32] as the model. We measure performance using the Intersection over Union (IOU) metric.

**Natural Language Processing**: Our research encompasses two crucial sub-fields within NLP: Text Classification and Machine Translation. For Text Classification, we focus on the SST2 [35] and the recent Ecommerce datasets [33]. We use the DistilBERT [34] model, fine-tuned for these tasks, with accuracy as our metric. Regarding Machine Translation, we utilize the Opus Books [36] dataset, specifically targeting English-French translation. For this task, the T5-Small [37] model is fine-tuned to assess its efficacy in translation and use the BLEU-Score to evaluate the model.

**Recommender Systems**: In Recommender Systems, our work spans Matrix Factorization (MF) and Click-Through Rate (CTR) prediction. We apply LightGCN [39] for MF, measuring performance with NDCG@10. For CTR prediction, we deploy DeepFM [40] and use AUC as the metric. Both tasks are executed on the MovieLens 1M [38] dataset.

**Tabular**: In the realm of tabular data, our research encompasses both regression and classification tasks. Specifically, we focus on a classification task involving water portability [41] and a regression task concerning house price [43] predictions. These tasks utilize datasets sourced from Kaggle, with careful consideration to avoid data leakage after the incorporation of GPT-based knowledge. For both tasks, we implement models based on the XGBoost [42] framework, known for its efficacy in handling structured data. For classification and regression tasks, we use the F1 Score and $R^2$ Score as the metrics, respectively.

**Graph Neural Networks**: In our Graph Neural Networks (GNN) research, we focus on two main objectives: node classification and link prediction. For node classification, we use the Cora dataset [44] and apply a Graph Convolutional Network (GCN) [45] as our model, evaluating performance based on accuracy. Additionally, for link prediction, we conduct experiments using the Pubmed dataset [44], employing a Variational Graph Autoencoder (VGAE) [46], with the AUC serving as the performance metric.

Table 2: Hyperparameter spaces of optimization tasks. We report the names, types, whether they are on a log scale, and the corresponding ranges of the hyperparameters for every task.

| Exp ID | Hyperparameter | Type | Log | Range |
|---|---|---|---|---|
| Image Classification | global_pool | cat | ✗ | [avg, max, avgmax, catavgmax] |
| | learning_rate | float | ✓ | $[10^{-5}, 10^{-1}]$ |
| | optimizer | cat | ✗ | [adam, sgd] |
| | epochs | int | ✗ | $[25, 200]$ |
| | weight_decay | float | ✓ | $[10^{-6}, 10^{-1}]$ |
| | dropout_rate | float | ✗ | $[0, 0.5]$ |
| | momentum | float | ✗ | $[0.5, 1]$ |
| | batch_size | ord | ✗ | $[32, 64, 128, 256, 512]$ |
| Image Segmentation | learning_rate | float | ✓ | $[10^{-5}, 10^{-1}]$ |
| | optimizer | cat | ✗ | [adam, sgd] |
| | epochs | int | ✗ | $[10, 50]$ |
| | weight_decay | float | ✓ | $[10^{-6}, 10^{-1}]$ |
| | activation | cat | ✗ | [relu, prelu] |

*Continued on next page*

| Exp ID | Hyperparameter | Type | Log | Range |
|---|---|---|---|---|
| | momentum | float | ✗ | $[0.5, 1]$ |
| | batch_size | ord | ✗ | $[4, 8, 16, 32, 64]$ |
| Text Classification | learning rate | float | ✓ | $[10^{-6}, 10^{-2}]$ |
| | epochs | int | ✗ | $[1, 4]$ |
| | dropout_rate | float | ✗ | $[0, 0.5]$ |
| | attention_dropout | float | ✗ | $[0, 0.5]$ |
| | seq_classif_dropout | float | ✗ | $[0, 0.5]$ |
| | batch_size | ord | ✗ | $[8, 16, 32, 64, 128]$ |
| | activation | cat | ✗ | [gelu, relu, silu] |
| | weight_decay | float | ✓ | $[10^{-6}, 0.1]$ |
| Translation | learning_rate | float | ✓ | $[10^{-6}, 10^{-2}]$ |
| | dropout | float | ✗ | $[0, 0.5]$ |
| | epochs | int | ✗ | $[1, 4]$ |
| | batch_size | ord | ✗ | $[16, 32, 64, 128]$ |
| | weight_decay | float | ✓ | $[10^{-6}, 0.1]$ |
| CTR | embedding_size | ord | ✗ | $[8, 16, 32, 64]$ |
| | learning_rate | float | ✓ | $[10^{-5}, 10^{-1}]$ |
| | optimizer | cat | ✗ | [adam, sgd] |
| | reg_weight | float | ✓ | $[10^{-6}, 10^{-1}]$ |
| | dropout_prob | float | ✗ | $[0, 0.5]$ |
| | batch_size | ord | ✗ | $[256, 512, 1024, 2048, 4096]$ |
| | mlp_hidden_size | ord | ✗ | $[32, 64, 128, 256, 512]$ |
| | num_mlp_layers | int | ✗ | $[1, 4]$ |
| MF | embedding_size | int | ✗ | $[16, 256]$ |
| | learning_rate | float | ✓ | $[10^{-5}, 10^{-1}]$ |
| | optimizer | cat | ✗ | [adam, sgd] |
| | reg_weight | float | ✓ | $[10^{-6}, 10^{-1}]$ |
| | batch_size | ord | ✗ | $[512, 1024, 2048, 4096]$ |
| | epochs | int | ✗ | $[100, 400]$ |
| | num_layers | int | ✗ | $[1, 5]$ |
| Tabular | max_depth | int | ✗ | $[3, 11]$ |
| | learning_rate | float | ✓ | $[10^{-3}, 1]$ |
| | min_child_weight | int | ✗ | $[1, 10]$ |
| | subsample | float | ✗ | $[0.5, 1]$ |
| | colsample_bytree | float | ✗ | $[0.5, 1]$ |
| | n_estimators | int | ✗ | $[100, 500]$ |
| | gamma | float | ✗ | $[0, 0.5]$ |
| | reg_alpha | float | ✗ | $[0, 1]$ |
| | reg_lambda | float | ✗ | $[0, 1]$ |
| | scale_pos_weight | float | ✗ | $[1, 10]$ |
| Node Classification | num_layers | int | ✗ | $[1, 5]$ |
| | learning_rate | float | ✓ | $[10^{-6}, 10^{-1}]$ |
| | optimizer | cat | ✗ | [adam, sgd] |
| | epochs | int | ✗ | $[1, 200]$ |
| | hidden_size | ord | ✗ | $[8, 16, 32, 64]$ |
| | activation | cat | ✗ | [relu, elu, silu] |
| | weight_decay | float | ✓ | $[10^{-6}, 10^{-1}]$ |
| | dropout | float | ✗ | $[0, 0.5]$ |
| Link Prediction | num_layers | int | ✗ | $[2, 5]$ |
| | learning_rate | float | ✓ | $[10^{-6}, 10^{-1}]$ |
| | optimizer | cat | ✗ | [adam, sgd] |
| | epochs | int | ✗ | $[1, 200]$ |

| Exp ID | Hyperparameter | Type | Log | Range |
|---|---|---|---|---|
| | hidden_channels | ord | ✗ | [16, 32, 64, 128, 256] |
| | out_channels | ord | ✗ | [16, 32, 64, 128, 256] |
| | activation | cat | ✗ | [relu, elu, silu] |
| | weight_decay | float | ✓ | $[10^{-6}, 10^{-1}]$ |
| | dropout | float | ✗ | [0, 0.5] |

Table 3: Performance comparison across 12 ML tasks among AgentHPO, Random Search, Bayesian optimization and Best Human Performance, with the highest **boldfaced** and second-highest underlined

| Exp ID | Trial | Random | Bayesian | GPT-3.5 | GPT-4 | Human |
|---|---|---|---|---|---|---|
| Image Classification Cifar-10 | 1 | $52.45_{\pm 23.64\%}$ | $48.36_{\pm 25.58\%}$ | $81.57_{\pm 1.22\%}$ | $\underline{82.59}_{\pm 1.09\%}$ | **85.05** |
| | 3 | $75.94_{\pm 4.38\%}$ | $74.17_{\pm 5.71\%}$ | $\underline{82.84}_{\pm 0.34\%}$ | $81.74_{\pm 0.72\%}$ | **85.05** |
| | 5 | $76.78_{\pm 4.85\%}$ | $77.57_{\pm 8.17\%}$ | $82.84_{\pm 0.34\%}$ | $\mathbf{85.08}_{\pm \mathbf{0.42}\%}$ | $\underline{85.05}$ |
| | 10 | $81.63_{\pm 3.88\%}$ | $79.87_{\pm 4.19\%}$ | $83.87_{\pm 1.18\%}$ | $\mathbf{85.18}_{\pm \mathbf{0.52}\%}$ | $\underline{85.05}$ |
| Image Classification Butterfly | 1 | $35.65_{\pm 29.86\%}$ | $\mathbf{20.32}_{\pm \mathbf{25.06}\%}$ | $81.51_{\pm 3.78\%}$ | $78.22_{\pm 0.56\%}$ | 82.74 |
| | 3 | $60.83_{\pm 21.73\%}$ | $\mathbf{25.55}_{\pm \mathbf{27.62}\%}$ | $\underline{83.97}_{\pm 2.12\%}$ | $83.67_{\pm 0.58\%}$ | 82.74 |
| | 5 | $73.47_{\pm 6.74\%}$ | $\underline{40.55}_{\pm 25.51\%}$ | $84.79_{\pm 1.01\%}$ | $\mathbf{85.20}_{\pm \mathbf{0.62}\%}$ | 82.74 |
| | 10 | $78.99_{\pm 3.22\%}$ | $\underline{63.57}_{\pm 8.67\%}$ | $84.79_{\pm 1.01\%}$ | $\mathbf{85.92}_{\pm \mathbf{0.57}\%}$ | 82.74 |
| Image Segmentation | 1 | $33.73_{\pm 24.89\%}$ | $31.87_{\pm 19.57\%}$ | $65.66_{\pm 4.33\%}$ | $\underline{69.30}_{\pm 0.83\%}$ | **70.83** |
| | 3 | $51.72_{\pm 18.60\%}$ | $49.82_{\pm 10.42\%}$ | $67.39_{\pm 3.88\%}$ | $\underline{69.42}_{\pm 0.69\%}$ | **70.83** |
| | 5 | $63.76_{\pm 5.61\%}$ | $56.23_{\pm 10.29\%}$ | $67.39_{\pm 3.88\%}$ | $\underline{70.04}_{\pm 0.57\%}$ | **70.83** |
| | 10 | $66.39_{\pm 4.58\%}$ | $63.06_{\pm 6.76\%}$ | $67.64_{\pm 4.00\%}$ | $\underline{70.04}_{\pm 0.57\%}$ | **70.83** |
| Text Classification SST2 | 1 | $75.90_{\pm 17.04\%}$ | $75.91_{\pm 16.18\%}$ | $89.79_{\pm 0.64\%}$ | $\underline{90.41}_{\pm 0.30\%}$ | **90.71** |
| | 3 | $83.88_{\pm 13.06\%}$ | $85.86_{\pm 5.09\%}$ | $90.02_{\pm 0.42\%}$ | $\underline{90.83}_{\pm 0.47\%}$ | **90.71** |
| | 5 | $89.45_{\pm 1.81\%}$ | $89.91_{\pm 1.00\%}$ | $90.02_{\pm 0.42\%}$ | $\mathbf{91.09}_{\pm \mathbf{0.43}\%}$ | $\underline{90.71}$ |
| | 10 | $90.28_{\pm 0.76\%}$ | $90.28_{\pm 0.45\%}$ | $90.34_{\pm 0.79\%}$ | $\mathbf{91.32}_{\pm \mathbf{0.11}\%}$ | $\underline{90.71}$ |
| Text Classification Ecommerce | 1 | $73.46_{\pm 26.12\%}$ | $79.44_{\pm 24.82\%}$ | $95.96_{\pm 0.72\%}$ | $\underline{97.44}_{\pm 0.18\%}$ | **97.53** |
| | 3 | $91.40_{\pm 14.26\%}$ | $96.05_{\pm 1.15\%}$ | $97.18_{\pm 0.53\%}$ | $\mathbf{97.70}_{\pm \mathbf{0.07}\%}$ | $\underline{97.53}$ |
| | 5 | $96.04_{\pm 1.84\%}$ | $96.68_{\pm 0.73\%}$ | $97.55_{\pm 0.36\%}$ | $\mathbf{97.77}_{\pm \mathbf{0.11}\%}$ | 97.53 |
| | 10 | $97.47_{\pm 0.18\%}$ | $97.21_{\pm 0.50\%}$ | $\underline{97.55}_{\pm 0.36\%}$ | $\mathbf{97.81}_{\pm \mathbf{0.13}\%}$ | 97.53 |
| Machine Translation | 1 | $18.26_{\pm 6.06\%}$ | $17.39_{\pm 6.85\%}$ | $17.41_{\pm 1.61\%}$ | $\underline{25.11}_{\pm 0.28\%}$ | **27.47** |
| | 3 | $20.58_{\pm 3.29\%}$ | $20.06_{\pm 3.57\%}$ | $21.32_{\pm 1.35\%}$ | $\underline{26.40}_{\pm 0.29\%}$ | **27.47** |
| | 5 | $23.68_{\pm 0.99\%}$ | $24.16_{\pm 2.98\%}$ | $21.53_{\pm 1.35\%}$ | $\mathbf{27.70}_{\pm \mathbf{0.45}\%}$ | $\underline{27.47}$ |
| | 10 | $25.72_{\pm 0.92\%}$ | $26.70_{\pm 0.59\%}$ | $22.43_{\pm 2.04\%}$ | $\mathbf{28.02}_{\pm \mathbf{0.61}\%}$ | $\underline{27.47}$ |
| MF | 1 | $13.16_{\pm 10.20\%}$ | $10.74_{\pm 10.06\%}$ | $26.21_{\pm 0.57\%}$ | $\underline{26.92}_{\pm 0.07\%}$ | **27.05** |
| | 3 | $17.33_{\pm 9.80\%}$ | $21.57_{\pm 3.03\%}$ | $26.57_{\pm 0.65\%}$ | $\mathbf{27.14}_{\pm \mathbf{0.04}\%}$ | $\underline{27.05}$ |
| | 5 | $23.81_{\pm 2.65\%}$ | $22.71_{\pm 2.99\%}$ | $27.05_{\pm 0.17\%}$ | $\mathbf{27.23}_{\pm \mathbf{0.05}\%}$ | $\underline{27.05}$ |
| | 10 | $26.02_{\pm 0.81\%}$ | $24.94_{\pm 1.57\%}$ | $\underline{27.13}_{\pm 0.13\%}$ | $\mathbf{27.24}_{\pm \mathbf{0.07}\%}$ | 27.05 |
| CTR | 1 | $71.26_{\pm 11.34\%}$ | $71.03_{\pm 10.32\%}$ | $81.68_{\pm 0.44\%}$ | $\underline{81.92}_{\pm 0.05\%}$ | **82.19** |
| | 3 | $76.98_{\pm 7.43\%}$ | $75.92_{\pm 6.79\%}$ | $\underline{82.05}_{\pm 0.14\%}$ | $82.01_{\pm 0.06\%}$ | **82.19** |
| | 5 | $78.85_{\pm 7.21\%}$ | $79.95_{\pm 3.80\%}$ | $\underline{82.14}_{\pm 0.06\%}$ | $82.06_{\pm 0.05\%}$ | **82.19** |
| | 10 | $81.94_{\pm 0.24\%}$ | $81.67_{\pm 0.92\%}$ | $\underline{82.14}_{\pm 0.06\%}$ | $82.09_{\pm 0.05\%}$ | **82.19** |
| Tabular Classification | 1 | $68.77_{\pm 2.27\%}$ | $68.71_{\pm 2.18\%}$ | $68.06_{\pm 0.11\%}$ | $\underline{69.33}_{\pm 0.12\%}$ | **72.3** |
| | 3 | $70.15_{\pm 1.29\%}$ | $70.58_{\pm 1.04\%}$ | $71.37_{\pm 0.32\%}$ | $\underline{71.57}_{\pm 0.17\%}$ | **72.3** |
| | 5 | $70.82_{\pm 1.25\%}$ | $70.95_{\pm 0.95\%}$ | $\underline{71.76}_{\pm 0.41\%}$ | $71.65_{\pm 0.19\%}$ | **72.3** |
| | 10 | $71.62_{\pm 0.55\%}$ | $71.58_{\pm 0.72\%}$ | $71.81_{\pm 0.37\%}$ | $\underline{72.01}_{\pm 0.35\%}$ | **72.3** |

| Exp ID | Trial | Random | Bayesian | GPT-3.5 | GPT-4 | Human |
|--------|-------|--------|----------|---------|-------|-------|
| Tabular Regression | 1 | $46.00_{\pm 12.55\%}$ | $50.75_{\pm 10.18\%}$ | $56.27_{\pm 0.35\%}$ | $56.57_{\pm 0.40\%}$ | **56.9** |
| | 3 | $54.90_{\pm 4.21\%}$ | $55.68_{\pm 2.23\%}$ | $56.55_{\pm 0.39\%}$ | $\mathbf{57.71}_{\pm \mathbf{0.05\%}}$ | $\underline{56.9}$ |
| | 5 | $56.12_{\pm 0.67\%}$ | $56.69_{\pm 0.23\%}$ | $56.76_{\pm 0.33\%}$ | $\mathbf{57.73}_{\pm \mathbf{0.07\%}}$ | $\underline{56.9}$ |
| | 10 | $56.49_{\pm 0.40\%}$ | $56.85_{\pm 0.05\%}$ | $56.78_{\pm 0.36\%}$ | $\mathbf{58.01}_{\pm \mathbf{0.11\%}}$ | $\underline{56.9}$ |
| Node Classification | 1 | $64.50_{\pm 19.18\%}$ | $55.57_{\pm 18.37\%}$ | $79.77_{\pm 0.71\%}$ | $80.06_{\pm 0.66\%}$ | **81.5** |
| | 3 | $76.85_{\pm 2.11\%}$ | $64.28_{\pm 14.52\%}$ | $80.60_{\pm 0.33\%}$ | $\underline{80.72_{\pm 0.27\%}}$ | **81.5** |
| | 5 | $77.64_{\pm 1.89\%}$ | $70.57_{\pm 11.43\%}$ | $80.93_{\pm 0.33\%}$ | $\underline{81.18_{\pm 0.45\%}}$ | **81.5** |
| | 10 | $78.80_{\pm 0.81\%}$ | $74.96_{\pm 2.70\%}$ | $81.13_{\pm 0.22\%}$ | $\underline{81.38_{\pm 0.22\%}}$ | **81.5** |
| Link Prediction | 1 | $90.21_{\pm 2.90\%}$ | $90.53_{\pm 2.40\%}$ | $90.02_{\pm 1.02\%}$ | $90.76_{\pm 0.47\%}$ | **95.12** |
| | 3 | $92.54_{\pm 2.07\%}$ | $92.36_{\pm 1.83\%}$ | $92.24_{\pm 1.43\%}$ | $\underline{92.51_{\pm 0.30\%}}$ | **95.12** |
| | 5 | $93.69_{\pm 1.65\%}$ | $93.33_{\pm 1.81\%}$ | $94.21_{\pm 0.55\%}$ | $\underline{94.87_{\pm 0.46\%}}$ | **95.12** |
| | 10 | $94.98_{\pm 0.73\%}$ | $94.84_{\pm 0.34\%}$ | $94.34_{\pm 0.65\%}$ | $\mathbf{95.39}_{\pm \mathbf{0.66\%}}$ | $\underline{95.12}$ |

## A.4. Prompts for AgentHPO

We here present the prompt template used to initialize the *Creator* and *Executor* agents.

## A.5. Creator Agent

---

**Creator Agent Prompts**

You are a task creation AI expert in machine learning that required to optimize the model's hyperparameter settings to accomplish the final objective. To achieve this, you need to check the previous hyperparameter tuning plan and completed tasks results. Based on this information, generate a new sub-task for the task execution agent that can solve the sub-task. Below is the basic information about the experimental settings:

{model_info}

{dataset_info}

Below are the hyper-parameters and corresponding candidates or values range that can be tuned for the task:

{hyperparameter_info}

To accomplish the task, you have access to the following tools:

Name: "LoadHistoricalTrainingLogs"
Description: "This tool is designed for easily loading and reviewing model training logs. It automatically accesses records of loss and accuracy metrics from different hyper-parameter settings."
Format your response as follows:
Objective: Define the final goal
Thought: Describe your reasoning process
Action: Specify the action to take; valid actions are 'Final Answer' or {tool_names}
Action Input: Input for the action
Observation: Outcome of the action
... (this Thought/Action/Action Input/Observation can repeat N times)
Thought: I now know the final answer
Final Answer: The proposed hyper-parameters for the task

---

Analyze the completed tasks and their outcomes. Propose a new task focused on unexplored hyperparameter spaces or optimization techniques to methodically reach the final objective. The task executor will adjust hyperparameters and run the training script. Ensure your proposed hyperparameters are distinct from those previously tested, and state your recommendation as the 'Final Answer'.

Objective: {optim_goal}
Thought: {agent_scratchpad}

## A.6. Executor Agent

**Executor Agent Prompts**

You are the machine learning experimenter and asked to finish the given objective below. To accomplish the task, you have access to the following tools:

Name: "LoadConfigs"
Description: "Useful for when you need to loading the model training configs and read the content. The file contains the hyper-parameters that used to define the training details of the model."

Name: "WriteConfigs"
Description: "Useful for when you need to writing the changed configs into file. Input should be the hyper-parameters that you want to write into the file IN JSON FORMAT. And you should also keep the unchanged Hyperparameter into the file."

Name: "ExecutePythonFile"
Description: "Useful for when you need to execute the python file to training the model"

Name: "LoadTrainingLogs"
Description: "Useful for when you need to loading the model training logs and read the content. The file contains the training logs (loss, accuracy) generated by training."

Use the following format:
Task: the input task you must solve
Thought: you should always think about what to do
Action: the action to take, should be one of [{tool_names}]
Action Input: the input to the action
Observation: the result of the action
... (this Thought/Action/Action Input/Observation can repeat N times)
Thought: I now know the final answer
Final Answer: the final answer to the original input question

After finish the task, analyze the training logs to make a summary about this experiment, including the analysis of the training trajectory and final training results. Then provide your answer with Final Answer.
Task: {task_name}
Thought:{agent_scratchpad}

## A.7. Example of Experimental Logs

In this section, we present selected experimental logs produced by AgentHPO for the image classification task using the Cifar dataset. Owing to space constraints, we illustrate only the initial experiment and the experiment that yielded the optimal results.

**A.7.1. GPT-3.5 Experimental Logs**



**GPT-3.5 Experimental Logs**

**Experiment 1**:
Based on the given information, I would recommend the following initial set of hyper-parameter settings to maximize accuracy:
global_pool: 'avgmax'
learning_rate: 1e-3
optimizer: 'adam'
epochs: 100
model_name: 'resnet18'
weight_decay: 1e-4
drop_rate: 0.3
momentum: 0.9
batch_size: 256

These settings aim to strike a balance between model complexity, regularization, and training efficiency, considering the dataset and the available models. Adjustments can be made based on the performance of the initial training runs.
Training Trajectory:
Epoch: [0, 5, 10, 15, 20, 25, 30, 35, 40, 45, 50, 55, 60, 65, 70]
Train Loss: [1.993, 0.9729, 0.5704, 0.2754, 0.0984, 0.0466, 0.0383, 0.0374, 0.0148, 0.0205, 0.0247, 0.0161, 0.0187, 0.018, 0.0113]
Train Acc: [27.47, 65.47, 80.205, 90.9475, 97.0475, 98.7675, 98.82, 98.7775, 99.615, 99.3625, 99.18, 99.5225, 99.3925, 99.4, 99.6625]
Val Loss: [1.6491, 0.9406, 0.6651, 0.6689, 0.7548, 0.962, 0.8462, 1.0437, 0.9182, 0.8397, 0.8824, 0.9271, 0.9714, 0.9869, 1.6386]
Val Acc: [42.57, 66.96, 77.0, 77.23, 78.53, 77.01, 79.4, 77.06, 80.11, 81.84, 80.69, 81.21, 80.82, 80.93, 75.28]
Total Training Time: 3096.64s
Final Validation Accuracy: 81.4500%
The final validation accuracy of the model trained with the given hyper-parameters is 81.45%.

**Experiment 7**:
The final answer of proposed new hyper-parameters is as follows:
global_pool: 'catavgmax'
learning_rate: 1e-4
optimizer: 'adam'
epochs: 150
model_name: 'resnet18d'
weight_decay: 1e-5
drop_rate: 0.4
momentum: 0.95
batch_size: 128

Training Trajectory:
Epoch: [0, 5, 10, 15, 20, 25, 30, 35, 40, 45, 50, 55, 60, 65, 70, 75]
Train Loss: [1.7289, 0.6375, 0.2236, 0.1073, 0.0614, 0.065, 0.0423, 0.0404, 0.036, 0.0518, 0.0569, 0.027, 0.0289, 0.0312, 0.0216, 0.0282]
Train Acc: [36.6725, 77.785, 92.315, 96.3375, 98.105, 97.865, 98.6375, 98.655, 98.74, 98.2775, 98.1325, 99.1125, 99.0875, 98.98, 99.3175, 99.1]
Val Loss: [1.5076, 0.7051, 0.6053, 0.9768, 0.8362, 0.7814, 0.8403, 0.7787, 0.805, 0.8523, 0.7458, 0.827, 0.945, 0.9369, 0.8229, 0.889]
Val Acc: [45.51, 75.41, 80.48, 76.1, 79.73, 80.63, 80.99, 82.31, 81.09, 81.14, 83.16, 82.28, 80.71, 80.78, 82.81, 81.17]
Total Training Time: 2816.78s
Final Validation Accuracy: 82.82%



The model trained with the recommended hyper-parameter settings achieved a final validation accuracy of 82.82%. The training trajectory indicates that the model may be overfitting as the training accuracy continues to increase while the validation accuracy fluctuates. Further adjustments to the hyper-parameters and potential regularization techniques may be necessary to improve the model's generalization performance.

**Final analyse result**:

1. Best Hyper-Parameter Found in Experiment:

The best hyper-parameter configuration was found in Experiment 7, which achieved a final test accuracy of 82.82%.

The hyper-parameter configuration is as follows:
- Global Pooling: 'catavgmax'
- Learning Rate: 1e-4
- Optimizer: 'adam'
- Epochs: 150
- Model Name: 'resnet18d'
- Weight Decay: 1e-5
- Drop Rate: 0.4
- Momentum: 0.95
- Batch Size: 128

2. Influence of Each Hyper-Parameter:
- Learning Rate: Lower learning rates (e.g., 1e-4) seem to perform better than higher rates. They help stabilize training and improve generalization.
- Optimizer: 'adam' consistently outperformed 'sgd' in terms of final test accuracy, indicating that it is a better choice for this task.
- Epochs: Longer training (e.g., 150 epochs) generally led to better performance, but potential overfitting after a certain point was observed.
- Model Name: 'resnet18d' was the model that consistently performed well across experiments.
- Weight Decay: Small weight decay values (e.g., 1e-5) performed better, suggesting the importance of regularization.
- Drop Rate: A drop rate of 0.4 seemed to be effective in preventing overfitting in several experiments.
- Momentum: Higher momentum values (e.g., 0.95) helped improve the model's training speed and performance in some cases.
- Batch Size: Larger batch sizes (e.g., 128) performed better, likely due to improved gradient estimates.

3. Potential Future Exploration Direction:
- Further exploration of learning rates: Experiment with a wider range of learning rates to find the optimal value for this specific task.
- Model architecture exploration: Consider exploring other architectures or variations of 'resnet18d' to see if any provide better results.
- Data augmentation: Investigate the impact of different data augmentation techniques on model performance to potentially improve generalization.
- Regularization methods: Experiment with different regularization techniques (e.g., dropout rates, weight decay) to find the right balance between overfitting and underfitting.
- Learning rate schedules: Implement learning rate schedules (e.g., learning rate annealing) to potentially enhance training stability.
- Hyperparameter tuning: Utilize automated hyperparameter optimization techniques (e.g., Bayesian optimization) to search for optimal hyperparameters efficiently.

### A.7.2. GPT-4 Experimental Logs

---

**GPT-4 Experimental Logs**

**Experiment 1**:
To tune the hyperparameters for the ResNet models on the CIFAR-10 dataset, it's important to select a range that has been generally found to work well for image classification tasks. Here is an initial suggestion for setting each hyperparameter, along with the reasoning behind each choice:
1. global_pool: Start with 'avg' as it's a common approach that helps to reduce overfitting compared to 'max' pooling.
2. learning_rate: Initialize in the middle of the log-scale range, around 1e-3. This is often a good starting point for learning rate. Be prepared to adjust this based on whether the loss is changing too slowly or too quickly.
3. optimizer: Use 'adam' to start with since it generally converges faster and with less tuning than 'sgd', though 'sgd' often yields better generalization with well-tuned learning rate and momentum.
4. epochs: Begin with a moderate number of epochs, such as 100. This represents a balance between underfitting (25 epochs might be too few) and overfitting or unnecessary computation (200 epochs could be too many if early stopping criteria or learning rate schedules are used).
5. model_name: Start with 'resnet18' as it is the most standard of the models and thus the most likely to have established baselines. After establishing a baseline with 'resnet18', you can experiment with the other variants.
6. weight_decay: Opt for a moderate value, like 1e-4. It helps regularize the learning process but setting it too high might lead to underfitting.
7. drop_rate: Begin with a small dropout rate, such as 0.1, to prevent overfitting, especially since CIFAR-10 is a relatively small dataset. Adjust as necessary based on validation set performance.
8. momentum: If you opt for 'sgd', set momentum around 0.9 as it is a typical value that helps accelerate the gradient vectors in the right directions, thus leading to faster converging.
9. batch_size: Start around 128 which is a balanced trade-off between the benefits of larger batch size (stability and performance) and smaller batch size (memory constraints and potential for better generalization).
Based on these considerations, here's a proposed initial configuration:
global_pool: avg,
learning_rate: 1e-3,
optimizer: adam,
epochs: 100,
model_name: resnet18,
weight_decay: 1e-4,
drop_rate: 0.1,
momentum: 0.9, # Only applicable for sgd optimizer
batch_size: 128

Please consider this as a starting point. It is crucial to use a systematic approach such as grid search, random search, or Bayesian optimization to find the best hyperparameters. Furthermore, proper techniques for avoiding overfitting, such as early stopping, data augmentation, or regularization, should complement this process. Monitoring both the training and validation loss will inform whether further adjustments are needed and help in the detection of overfitting or underfitting.
Training Trajectory:
Epoch: [0, 5, 10, 15, 20, 25, 30, 35, 40, 45]
Train Loss: [1.5783, 0.5534, 0.2122, 0.1135, 0.0776, 0.0759, 0.054, 0.0521, 0.0536, 0.0508]
Train Acc: [42.0275, 80.615, 92.675, 96.1475, 97.3, 97.44, 98.1325, 98.2725, 98.1925, 98.2525]
Val Loss: [1.7791, 0.7139, 0.7447, 0.8525, 0.7293, 0.9376, 0.8194, 0.8674, 0.8768, 0.9351]
Val Acc: [39.43, 75.77, 78.15, 78.3, 81.33, 78.63, 80.68, 80.24, 80.23, 78.59] Total Training Time: 1438.35s

Final Validation Accuracy: 80.4100%

The initial hyperparameter configuration led to a final validation accuracy of 80.41% on the CIFAR-10 dataset with the ResNet18 model. However, the model shows signs of overfitting as evidenced by the high training accuracy and the increase in validation loss over epochs. Further hyperparameter tuning and regularization techniques are necessary to improve the model's generalization to new data.

**Experiment 4**:

The proposed new hyperparameters for the next experiment are:

global_pool: catavgmax, # A combination of 'avg' and 'max' pooling not yet tested learning_rate: 1e-4, # A value that has shown promise in Experiment 3

optimizer: adam, # 'adam' is chosen for fast convergence

epochs: 150, # An increased number of epochs to allow learning rate scheduler to work effectively

model_name: resnet18d, # A variant of ResNet18 that hasn't been tested yet

weight_decay: 2e-4, # A moderate value for weight decay, higher than in Experiment 1 but lower than in Experiment 2

drop_rate: 0.25, # A value between the drop rates tested in Experiments 2 and 3

momentum: 0.9, # Momentum is not applicable for 'adam' optimizer

batch_size: 96, # A batch size between those tested in Experiments 1 and 2

This configuration is chosen to explore different areas of the hyperparameter space that have not been fully tested in the previous experiments, with a focus on improving model generalization and reducing overfitting.

Training Trajectory:

Epoch: [0, 5, 10, 15, 20, 25, 30, 35, 40, 45, 50]

Train Loss: [1.865, 0.7766, 0.3374, 0.1026, 0.0508, 0.043, 0.0444, 0.0298, 0.0285, 0.028, 0.0266]

Train Acc: [31.9975, 72.8975, 88.9175, 97.035, 98.56, 98.755, 98.6025, 99.0475, 99.0875, 99.1025, 99.1675]

Val Loss: [1.5616, 0.7803, 0.6193, 0.7015, 0.7287, 0.6634, 0.7635, 0.9808, 0.7534, 0.8566, 0.8475]

Val Acc: [45.55, 72.9, 78.83, 78.98, 79.6, 82.44, 80.86, 77.17, 82.24, 79.39, 80.99]

Total Training Time: 2585.06s

Final Validation Accuracy: 82.4500%

The experiment with the specified configuration achieved a high training accuracy of 99.1675% but a moderate validation accuracy of 82.45%, indicating potential overfitting. Future work should focus on improving generalization to avoid overfitting and enhance performance on unseen data.

**Final analyse result**:

1. Best Hyperparameter Configuration Found in Experiments:

- Experiment 4 yielded the highest final validation accuracy (82.45%). The key hyperparameters in this experiment were:
- Global Pool: catavgmax
- Learning Rate: 1e-4
- Optimizer: adam
- Epochs: 150
- Model Name: resnet18d
- Weight Decay: 2e-4
- Drop Rate: 0.25
- Batch Size: 96

2. Influence of Each Hyperparameter:

- Global Pool: Switching between 'avg', 'max', and combined strategies showed varying impacts on overfitting and model performance.
- Learning Rate: Lower learning rates (1e-4 to 5e-5) generally improved generalization but required more epochs for convergence.
- Optimizer: 'adam' provided fast convergence but sometimes led to overfitting; 'sgd' improved generalization but required careful tuning of learning rate and momentum.
- Epochs: Higher epoch counts (150-200) were beneficial for models with lower learning rates, allowing for more gradual learning and convergence.

- Model Variants: Different ResNet variants (e.g., 'seresnet18', 'skresnet18') influenced the model's ability to learn and generalize, with each showing unique performance characteristics.
- Weight Decay: Higher values (up to 5e-4) helped in regularizing the model and reducing overfitting.
- Drop Rate: Varied impacts were observed; higher drop rates (0.3-0.4) seemed beneficial in some cases for preventing overfitting.
- Momentum: Used with 'sgd', values around 0.85 to 0.95 helped in smoothing out updates and improving convergence.
- Batch Size: Middle-range sizes (64-128) provided a balance between performance and generalization.

3. Potential Future Exploration Directions:
- Hyperparameter Ranges: Further fine-tuning within the established ranges could yield improvements, especially balancing learning rate and epochs for different optimizers.
- Regularization Techniques: Exploring additional regularization methods (e.g., more advanced dropout strategies, data augmentation) could help in reducing overfitting.
- Model Variants: Testing other ResNet variants or even different architectures might reveal models better suited to the CIFAR-10 dataset.
- Pooling Strategies: Continued exploration of different global pooling methods (like combinations of 'avg' and 'max') may uncover optimal feature extraction techniques.
- Learning Rate Schedulers: Experimenting with different schedulers could lead to better training dynamics.
- Batch Size Optimization: Investigating the impact of batch size on model performance and generalization in more detail could be valuable, as different sizes may affect the noise in gradient estimates.
- Ensemble Techniques: Combining predictions from models trained with different hyperparameters might enhance overall performance.

In summary, while Experiment 4 provided the best results, there is room for improvement in generalization and performance. Further experiments should focus on fine-tuning hyperparameters, exploring new regularization techniques, and possibly trying different model architectures or ensemble methods.

