# OpenReview forum: "AgentHPO: Large Language Model Agent for  Hyper-Parameter Optimization"
_CPAL.cc/2025/Proceedings_Track — CPAL 2025 (Proceedings Track) Poster_

### Official Review · Reviewer_QgHN · 2025-01-03

**Rating:** 8
**Confidence:** 4

**Review:**

This paper introduces AgentHPO, a novel framework for hyperparameter optimization (HPO) that leverages LLM-based autonomous agents in a collaborative creator-executor structure. AgentHPO achieves comparable, and often superior, performance to human experts and baseline methods across a diverse set of 12 machine learning tasks. The framework significantly improves trial efficiency, simplifies setup complexity, and enhances interpretability compared to traditional AutoML approaches. The empirical results show AgentHPO's practical viability in real-world scenarios, are further strengthened by a detailed analysis of the agents' optimization strategies.

The reviewer finds the paper super interesting and believes it is a suitable topic for CPAL.

---

### Official Review · Reviewer_7dGw · 2025-01-07
**review for submission 92**

**Rating:** 6
**Confidence:** 3

**Review:**

This paper introduces a novel system for automating the process of hyperparameter optimization (HPO) in machine learning models using large language models (LLMs). Named AgentHPO, this system utilizes LLMs to process task-specific information and iteratively optimize model hyperparameters based on historical data. The approach promises reduced trial numbers, simpler setup, and increased interpretability compared to traditional Automated Machine Learning (AutoML) systems.

[Strengths]:
1. Innovative Approach: The paper leverages LLMs to automate and optimize the HPO process, which is a cutting-edge approach combining the fields of machine learning and natural language processing.
2. Empirical Validation: Extensive experiments across diverse machine learning tasks demonstrate that AgentHPO matches or surpasses the performance of the best human optimizations while reducing the number of required trials and simplifying the setup process.

[Drawbacks]:
1. Dependence on LLMs' Knowledge Base: The performance of the AgentHPO system heavily relies on the knowledge encoded in LLMs, which may limit its effectiveness to the domains and data the models have been exposed to during training.
2. Potential for Overfitting and Under-generalization: There is a risk of overfitting to the kinds of datasets and problems that LLMs are more familiar with, potentially leading to under-generalization when faced with significantly different tasks or data distributions.

---

### Official Review · Reviewer_Show · 2025-01-12

**Rating:** 6
**Confidence:** 4

**Review:**

This paper introduces AgentHPO, a novel hyperparameter optimization framework powered by LLMs. The approach uses an autonomous "Creator-Executor" agent framework, where the Creator generates initial hyperparameter settings based on natural language input and historical experiment logs, while the Executor carries out model training, logging results, and refining future suggestions. By simulating a human expert’s decision-making process, AgentHPO improves trial efficiency, simplifies HPO configuration, and provides interpretable optimization results. Empirical evaluations across 12 machine learning tasks demonstrate AgentHPO’s superior performance compared to traditional AutoML methods and even human benchmarks, particularly when using GPT-4. The analysis highlights AgentHPO's adaptability to unseen tasks and datasets, showcasing its potential to enhance interpretability and resource efficiency in HPO.

Strengths:
1. The Creator-Executor setup introduces a structured and interpretable approach to HPO, effectively emulating expert-like decision-making.
2. The method is validated on 12 diverse ML tasks, demonstrating consistent improvements over traditional AutoML, Bayesian optimization, and random search.
3. AgentHPO provides detailed textual explanations of hyperparameter choices, enhancing transparency and user trust.
4. Achieves strong performance within early trials due to leveraging pre-trained LLMs, particularly GPT-4’s superior reasoning capabilities.

Weaknesses:
1. I would suggest the authors to report the expenses or financial efficiency of the proposed method. The hard reliance on large LLMs, especially GPT-4, increases operational costs, potentially limiting real-world usability for resource-constrained users.
2. Although OPRO is used as a baseline, additional comparisons with recent advanced AutoML frameworks (e.g., Optuna or BOHB) would strengthen the evaluation.

The paper presents improves hyperparameter optimization by integrating LLM-based agents. The proposed framework is methodically sound and demonstrates strong empirical performance across multiple tasks. However, addressing computational efficiency and comparing with more diverse baselines would enhance its practical impact.

---

### Meta-Review · Area_Chair_aBgq · 2025-02-04

**Recommendation:** Accept (Poster)
**Confidence:** 3

**Metareview:**

The paper introduces AgentHPO, a hyperparameter optimization framework leveraging LLM-based autonomous agents in a Creator-Executor structure. It demonstrates strong empirical performance across 12 machine learning tasks, surpassing traditional AutoML methods and human benchmarks, with enhanced interpretability and trial efficiency. While addressing computational costs and including comparisons with more diverse baselines would further strengthen its impact, the paper makes an interesting contribution to hyperparameter optimization, and worths acceptance.

---

### Decision · Program_Chairs · 2025-02-11

Accept (Poster)